# A Conceptual Model for Forest Naturalness Assessment and Application in Quebec's Boreal Forest

**Sylvie Côté [1],\*, Louis Bélanger [1] , Robert Beauregard [1], Évelyne Thiffault [1] and Manuele Margni [2] **

[1]  Department of Wood and Forestry Sciences, Université Laval, Quebec City, QC G1V 0A6, Canada; Louis.Belanger@sbf.ulaval.ca (L.B.); Robert.Beauregard@sbf.ulaval.ca (R.B.); Evelyne.Thiffault@sbf.ulaval.ca (É.T.)
[2]  CIRAIG, Polytechnique Montréal, Department of Mathematical and Industrial Engineering, Montreal, QC H3C 3A7, Canada; manuele.margni@polymtl.ca
\*  Correspondence: sylvie.cote.14@ulaval.ca; Tel.: +1-418-424-0422

**Abstract:** *Research Highlights:* To inform eco-designers in green building conception, we propose a conceptual model for the assessment of the impact of using wood on the quality of ecosystems. *Background and Objectives:* The proposed model allows the assessment of the quality of ecosystems at the landscape level based on the condition of the forest and the proportion of different practices to characterize precisely the forest management strategy. The evaluation provides a numerical index, which corresponds to a suitable format to inform decision-making support tools, such as life cycle analysis. *Materials and Methods:* Based on the concept of naturalness, the methodology considers five naturalness characteristics (landscape context, forest composition, structure, dead wood, and regeneration process) and relies on forest inventory maps and data. An area within the boreal black spruce-feathermoss ecological domain of Quebec (Canada) was used as a case study for the development of the methodology, designed to be easily exportable. *Results:* In 2012, the test area had a near-natural class (naturalness index NI = 0.717). Simulation of different management strategies over 70 years shows that, considering 17.9% of strict protected areas, the naturalness index would have lost one to two classes of naturalness (out of five classes), depending on the strategy applied for the regeneration ($0.206 \leq \Delta NI \leq 0.413$). Without the preservation of the protected areas, the management strategies would have further reduced the naturalness ($0.274 \leq \Delta NI \leq 0.492$). Apart from exotic species plantation, the most sensitive variables are the percentage of area in irregular, old, and closed forests at time zero and the percentage of area in closed forests, late successional species groups, and modified wetlands after 70 years. *Conclusions:* Despite the necessity of further model and parameter validation, the use of the index makes it possible to combine the effects of different forestry management strategies and practices into one alteration gradient.

**Keywords:** naturalness; forest management intensity; land use intensity; quality of ecosystems; boreal forest

---

## 1. Introduction

Quantitative tools to discriminate between different wood supplies depending on forest management and wood procurement practices are needed to inform architects and designers planning the eco-design of buildings. Using the science of applied ecology, such tools should make it possible to evaluate and compare the impact of different forestry strategies and the combination of practices on the quality of forest ecosystems.

This study aims to develop a methodology to characterize the potential impacts on ecosystem quality of different forestry management practices, in the perspective of describing the intensity of

land use as driven by forestry. The methodology is based on the naturalness concept and relies on forest inventory maps and data. Our methodology allows the evaluation of combinations of practices and provides one numerical index, a suitable format for further use in decision-making support tools for eco-design and green building conception, such as life cycle analysis (LCA) [1]. An area within the boreal black spruce ecological domain of Quebec (Canada) was used as a case study for the development of the methodology.

The specific objectives of the study are to:

(1) Develop a naturalness evaluation model using the example of the boreal black spruce-feathermoss ecological bioclimatic domain of Quebec (Canada).
(2) Apply the model over time on three forest management units (3 FMU) to analyze the variability of the naturalness evaluation associated with changes in forest management strategies and practices.
(3) Perform a sensitivity analysis of the model to (hypothetical) high pressure levels and to identify the most sensitive variables.

The need for evaluating the quality of ecosystems in relation with their anthropic uses presents many challenges. As land use, and particularly land use change, is one of the main drivers of biodiversity loss [2], there is a desire to express its impact on the quality of ecosystems in terms of biodiversity damage in LCA [3,4]. The latest proposed LCA approach uses potential species loss from land use as an indicator; for forestry, it considers two land use intensities (intensive and extensive) [3]. This proposal raises two issues: Biodiversity data and indicators' availability, and land use intensity evaluation. Concerning biodiversity, potential species loss is still proposed as the biodiversity indicator even if it does not reflect the multidimensional character of biodiversity and might lead to inappropriate conclusions [5]. As stated by Souza et al. [6], the biodiversity models proposed up to now do not grasp the full reach of the phenomena involved, such as functional effects and impacts on populations. Furthermore, there are data gaps in biodiversity: Biodiversity data are often fragmentary (they do not include all taxa) and of varying quality (all biomes are not evenly studied, especially the boreal biome for which data are particularly scarce). For instance, boreal forests are underrepresented in global biodiversity databases (see GLOBIO [7]; PREDICTS [8]). Concerning the intensity, forest management strategies generally include a mix of practices that have different impacts on the ecosystem, and the intensity is related to the recurrence of treatments over the same area planned in the silvicultural scenario. Because of these issues, we propose an alternative approach to evaluate ecosystem quality related to forest management, one that focuses on habitat characteristics and the concept of naturalness.

Many authors have proposed to use the concepts of naturalness and hemeroby in impact evaluation of land use (such as forestry) on the quality of ecosystems in LCA [9–13]. Naturalness is defined as "*the similarity of a current ecosystem state to its natural state*" [14], whereas hemeroby expresses "*distance to nature*" in landscape ecology [11]. The use of these concepts can provide a management guide that overcomes the challenge of data gaps in biodiversity. Even if the concepts of naturalness and hemeroby are closely related, one is not the exact inverse of the other. There is also divergence concerning the highest degree of alteration that should be included [14]. To clarify, we associate the naturalness concept with forest ecosystems, as shown in Figure 1; its lower class, i.e., the most altered state, corresponds to artificial forests [15,16] created by humans and showing deep modifications to the ecosystem and its species composition [15]. On the other hand, in the hemeroby scale, the alteration gradient is further developed and extended to sealed soils, and constructed, degraded, or devastated areas [11], with some authors even distinguishing dumpsites and partially built areas from sealed soils [9]. As stated by Winter [14], "*greater naturalness is characterized by a large number of adapted, specialized and often endangered plant and animal species*". Thus, in order to prevent or limit forest biodiversity loss due to forestry, the emphasis should be put on maintaining or restoring a high degree of naturalness. The concept of naturalness is well adapted to evaluate forestry management practices, but its application to evaluate the full alteration range of different land uses beyond forestry will require further work for proper insertion in the hemeroby concept that addresses a larger alteration gradient. Since the scope of

this paper focuses on the impacts of forest management practices, the evaluation is restricted to the naturalness part of the alteration gradient.

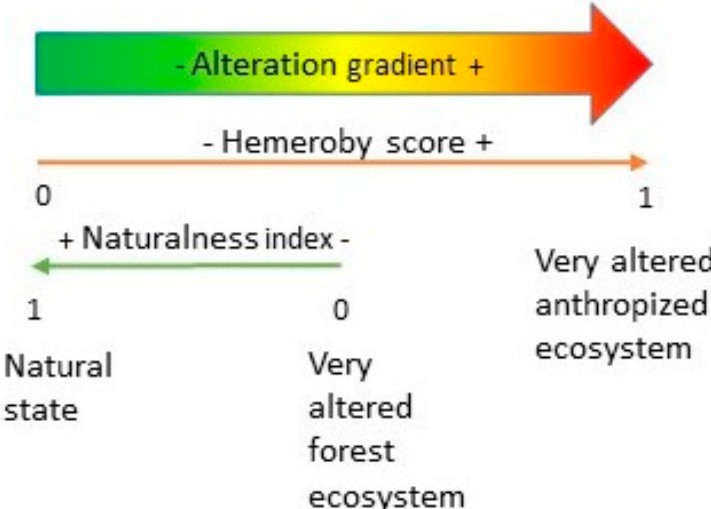

**Figure 1.** Naturalness and hemeroby along the alteration gradient (adapted from Winter et al. [17]).

The use of an index to evaluate the departure from the natural state along the alteration gradient avoids the problem of multiple classification of naturalness encountered in the literature [18] and allows inclusion of realistic forest management strategies, which involve a mix of different practices. Naturalness evaluation based on habitat characteristics is possible with actual data, and the future evolution of characteristics can be predicted. However, impacts on biodiversity are more challenging to assess considering the need of biodiversity indicators that encompass multidimensional characteristics of the biodiversity concept [5] and the uncertainty related to the timelag existing between habitat destruction and species extinction or extirpation [19].

Forest naturalness can be defined and evaluated using three interdependent approaches based on structure (i.e., spatial arrangement of the various components of the ecosystem [20]), composition, and processes [21]. Generally, naturalness assessment results from the comparison between the actual condition and a reference state [22] using either historical inventory data, prior to commercial forest exploitation, or modelling studies of forest dynamics evaluating the range of natural variability [23]. Where historic data are not available, the reference state, which corresponds to the most natural state, can be associated with a position along the alteration gradient [17]. Many methods have been proposed to assess naturalness [14], which is coherent with the fact that the choice of variables for naturalness studies must be adapted to regional conditions and knowledge [17].

The method developed here uses condition and pressure indicators. Indicators correspond to specific "elements of the forest system (e.g., species, processes and habitats) that correlate with many other unmeasured elements of the system" [24]. Condition or state indicators describe the current status or condition of a characteristic; pressure indicators represent the level of a pressure that affects the condition of a characteristic (i.e., an action that is causing the condition to degrade or improve) [24]. Thus, condition indicators are related to the concept of naturalness (i.e., the similarity of a current ecosystem state to its natural state), whereas pressure indicators are rather related to the hemeroby concept (i.e., distance to nature). However, pressure indicators can still be used to evaluate naturalness considering their effects on the condition of a characteristic.

We developed our conceptual model for naturalness assessment at the landscape level in a way that it could be easily adapted to other contexts and available data. Our method explores the application of non-linear relationships to integrate the notion of ecological thresholds in the naturalness assessment; habitat thresholds correspond to points or zones at which relatively rapid changes occur from one

ecological condition to another [25]. We also propose an original method for handling condition and pressure indicators in the index calculation.

## 2. Materials and Methods

### 2.1. Conceptual Model

Designing a model to feed decision support systems relying on science-based evidence requires condensing and summarizing original information from studies and reviews in a form accessible to decision-makers [26]. This challenging exercise involves a choice of critical criteria relevant to the decision; in this case, assessing the impact of forest management practices on ecosystem quality.

The model we propose determines an aggregated naturalness index (NI) based on five forest naturalness characteristics: (1) Landscape context, (2) composition, (3) structure, (4) dead wood (DW), and (5) regeneration process (RP). The landscape context characteristic refers to forest habitat at the landscape level; composition corresponds to tree species composition; structure considers age structure as well as physical vertical and horizontal structure; dead wood focuses on coarse woody debris; finally, the regeneration process characteristic refers to the forest renewal mode (see Appendix A for more details about indicators and measures for each characteristic).

The conceptual model developed for naturalness assessment in the black spruce and feathermoss domain of Quebec's boreal forest is presented in Figure 2.

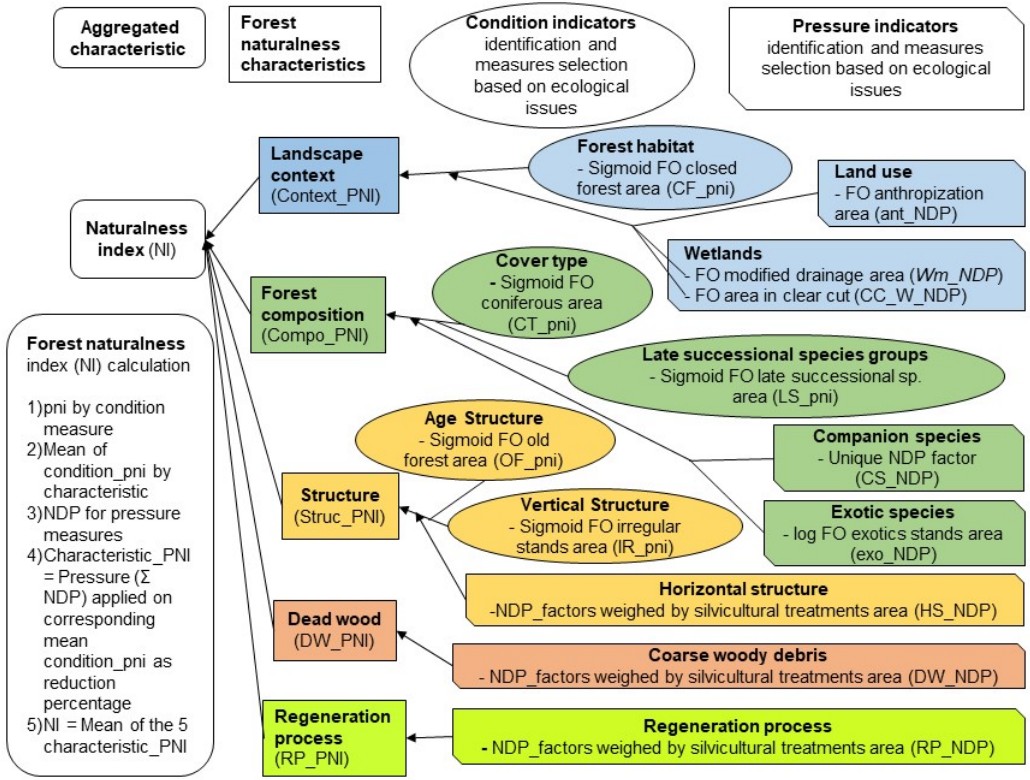

**Figure 2.** A naturalness evaluation conceptual model for the black spruce and feathermoss boreal forest. PNI: Partial naturalness index for naturalness characteristic; pni: partial naturalness index for condition indicator; NDP: Naturalness degradation potential; FO: Function of.

For each naturalness characteristic, the selection of condition indicators was based on ecological issues relevant to this region [27]. Corresponding measures for each indicator (Table 1) were identified from available ecoforest maps and from relevant modelling studies.

**Table 1.** Naturalness characteristics, indicators and measures.

| Naturalness Characteristic | Condition Indicator(s) | Condition Measure(s) | Pressure Indicator(s) | Pressure Measure(s) |
|---|---|---|---|---|
| Landscape context (Context_PNI [1]) | Forest habitat (closed forest) (CF_pni) | % of terrestrial area of forest >40 years old (CF) | Land use (ant_NDP) | % of terrestrial area with anthropization (ant) |
| | | | Wetlands (Wm_NDP) | % of modified wetlands (Wm) |
| | | | (W_CC_NDP) | % of humid area in clear cut (W_CC) |
| Forest composition (Compo_PNI) | Cover type (CT_pni) | % of forest area with coniferous cover type (CT) | Companion species (CS_NDP) | Recognized companion species diminution (CS) |
| | Late successional species (LS_pni) | % of forest area in late successional species groups (LS) | Exotic species (exo_NDP) | % of forest area of exotic species stands (exo) |
| Structure (Struc_PNI) | Age structure (OF_pni) | % of forest area of old forests (>100 years old) (OF) | Horizontal structure (HS_NDP) | HS NDP_factor by silvicultural treatment weighed by % of forest area |
| | Vertical structure (IR_pni) | % of forest area of irregular forests (IR) | | |
| Dead wood (DW_PNI) | | | Coarse woody debris (DW_NDP) | DW NDP_factor by silvicultural treatment weighed by % of forest area |
| Regeneration process (RP_PNI) | | | Regeneration process (RP_NDP) | RP NDP_factor by silvicultural treatment weighed by % of forest area |

[1] PNI: Partial naturalness index for naturalness characteristic; pni: Partial naturalness index for condition indicator; NDP: Naturalness degradation potential.

First, the measures of condition are used to evaluate partial naturalness indexes (PNI/pni) using a sigmoidal curve (Figure 3a). Measures of pressure are then used to evaluate naturalness degradation potentials (NDP), using either linear or logarithmic curves (Figure 3b,c) or territory specific NDP factors related to practices weighed by the percentage of area as described in the section test area. Then, for each naturalness characteristic, i, the partial naturalness index (Characteristic_PNI$_i$) is calculated as follows (see Table 2 for the calculation details of each characteristic):

$$\text{Characteristic\_PNI}_i = \left(\frac{1}{n}\sum\nolimits_{j=1}^{n}\text{Condition\_pni}_j\right) \times \left(1 - \sum\nolimits_{k=1}^{m}\text{NDP}_k\right) \tag{1}$$

where PNI/pni = partial naturalness index; NDP = naturalness degradation potential; n = number of condition indicators, j, for each characteristic, i (up to two); m: number of NDP, k, for each characteristic, i.

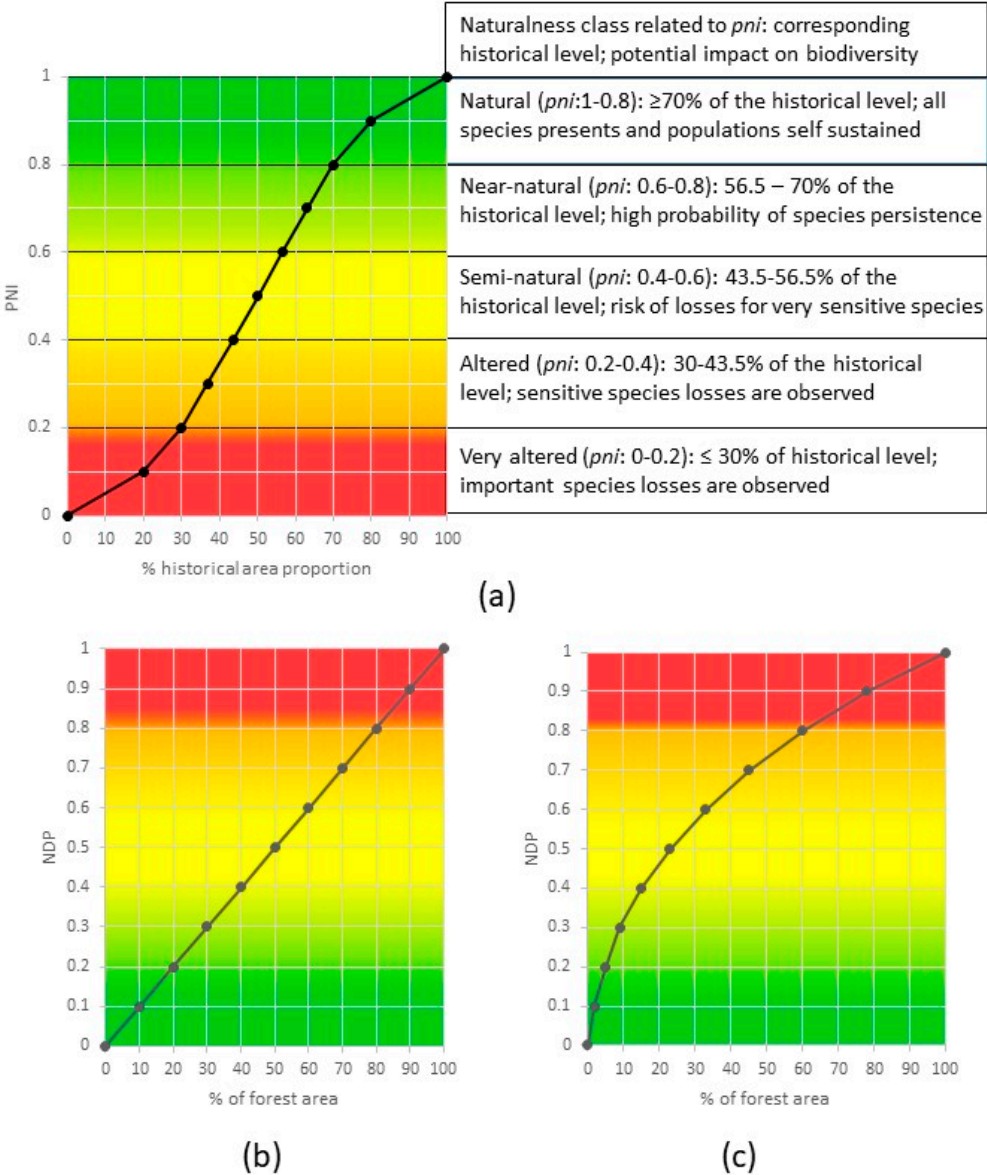

**Figure 3.** Generic curves used for modelling. (**a**) Sigmoid curve for the partial naturalness index of condition indicators. (**b**) Linear curve for naturalness degradation potential evaluation for pressure indicators proportional to the area. (**c**) Logarithmic curve for naturalness degradation potential evaluation for high potential impact pressure indicators.

**Table 2.** Characteristic_PNI equations for each naturalness characteristic.

| Naturalness Characteristic | Characteristic_PNI [1] Equation |
|---|---|
| Landscape context | $Context\_PNI = CF\_pni \times (1 - (ant\_NDP + Wm\_NDP + W\_CC\_NDP))$ |
| Forest Composition | $Compo\_PNI = ((CT\_pni + LS\_pni)/2) \times (1 - (exo\_NDP + CS\_NDP))$ |
| Structure | $Struc\_PNI = ((OF\_pni + IR\_pni)/2) \times (1 - HS\_NDP)$ |
| Dead wood | $DW\_PNI = 1 - DW\_NDP$ |
| Regeneration process | $RP\_PNI = 1 - RP\_NDP$ |

[1] PNI/pni: partial naturalness index; NDP: naturalness degradation potential; see Table 1 for variables definitions.

The naturalness index and the partial naturalness indexes of both levels (characteristic and condition) range from 1 (natural) to 0 (very altered). To ease the interpretation and the discussion, we divided this range in five equal classes with an associated colour code: Natural (dark green): 1–0.8;

near natural (light green): 0.799–0.6; semi-natural (yellow): 0.599–0.4; altered (orange): 0.399–0.2; very altered (red): 0.199–0 (see Figure 3a).

The evaluation of a condition indicator is based on a comparison with historical values. For each condition indicator, a partial naturalness index (condition_pni) is evaluated using a sigmoid curve relating the measure, corresponding to the actual proportion of the historical area percentage (or at a given time in the future, for forecasting scenarios) to the pni (Figure 3a). The sigmoid curve is considered a good representation of the type of relationship existing between the amount of habitat and species' response [28] and reflects the presence of thresholds. In our approach, habitat thresholds were used to determine changes between naturalness classes in order to put in relation the degree of ecosystem alteration and its potential effect on biodiversity. Many authors reported an important reduction in biodiversity when the amount of habitat is below 30% of the historical level [25,29,30]; hence, the upper limit of the "very altered" class was set at 30% of the historical level. As naturalness classes correspond to equal divisions of the alteration gradient, the sigmoid curve was centered at 50% of the historical level. Therefore, on the opposite side of the gradient, the lower limit of the "natural" class was set at 70% of the historical level for each condition indicator; the lower and upper limits for the "semi-natural" class were therefore set at 43.5% and 56.5% of the historical level. This range corresponds roughly to the range of the mean thresholds (absence, colonization, extinction, and persistence) for the amount of habitat of 44% to 61% of historical levels observed for breeding birds by Zuckerberg and Porter [28]. The upper limit of the "altered" class is slightly above the threshold of 40% of the historical level values observed for the persistence of some specialized species [31]. Therefore, we can consider that the probability of persistence of a species is generally high in the "near-natural" class; it then declines in the "semi-natural" class as some very specialized species might be affected. Sensitive species will be lost in the "altered" class and many species will be lost in the "very altered" class (Figure 3a). PNIs and exo_NDP was evaluated by linear interpolation between curve points (see Supplementary Materials Table S1: PNIs and exo_NDP determination).

Measures of pressures are used to evaluate a naturalness degradation potential (NDP) [9]. In the model, the total naturalness degradation (ΣNDP) for each naturalness characteristic was applied as a relative reduction (expressed in the percent of naturalness degradation) of the corresponding mean of condition_pni (Formula 1). Higher values of NDP represent a higher potential of naturalness degradation, corresponding to the red class.

There are four possible approaches to determine an NDP. The first one considers a unique degradation factor for the whole area. The second considers that NDP is proportional to the area under pressure using a linear relationship (Figure 3b). The third approach relates to potentially high impact interventions, and considers that NDP is evaluated using a logarithmic curve (Figure 3c). A practice is considered to have a high potential impact when a small proportion of impacted area can have detrimental effects over a wider area. For example, if the proportion of exotic species stands reaches 60%, this corresponds to a very high potential of naturalness degradation resulting from the modification of the forest matrix. The fourth approach for NDP evaluation is used for variables that cannot be measured or derived from a forest cartography or inventory (i.e., horizontal structure (HS), dead wood (DW), and regeneration process (RP)). For each of these variables, the pressure level associated with silvicultural treatments was rated to reflect the effect of the disturbance intensity on the variable considered, using degradation factors based either on data (for dead wood) or on expert opinion. Expert opinion is often used in decision support systems applied to environmental management either to compensate for the lack of data or to interpret scientific results in order to provide guidelines based on science [26]. The evaluation of NDP factors related to the fourth evaluation approach (see examples in Table 3, Table 4, and Table 5) could be further developed using participatory methods, such as the Delphi survey [32], involving a team of professionals. For dead wood and regeneration process, to overcome the absence of condition measures, the NDP is applied on the value corresponding to the natural state (condition_pni = 1); therefore, in these cases: Characteristic_PNI = 1 − NDP.

**Table 3.** Naturalness degradation potentials (NDPs) for long lived companion species.

| Long Lived Companion Species Status | NDP_Factors |
|---|---|
| Recognized species in diminution | 0.2 |
| Recognized extirpated species (theoretical) | 0.6 |

**Table 4.** Naturalness degradation potential (NDP) for horizontal structure (HS) by silvicultural treatments in Quebec's boreal forest.

| Practice | NDP_Factors | % Forest_Area | NDPx |
|---|---|---|---|
| Plantation—thinning | 1 | 0.47% | 0.0047 |
| Plantation | 0.9 | 4.62% | 0.0416 |
| Thinning (natural), strip cutting | 0.8 | 0.46% | 0.0037 |
| Precommercial thinning (natural), release | 0.75 | 1.68% | 0.0126 |
| Salvage logging | 0.6 | 0.07% | 0.0004 |
| Careful logging (CL) and clear cut | 0.35 | 13.93% | 0.0488 |
| CLASS, variable retention cut (2% vol) | 0.3 | 0.00% | 0.0000 |
| Partial cutting | 0.2 | 0.12% | 0.0002 |
| Undisturbed or natural disturbances | 0 | 78.65% | 0.0000 |
| Actual HS_NDP | | | 0.1120 |

Note: NDP_factors: naturalness degradation potential factors related to practices; % for_area: percentage of forested area (in 2012); NDPx: Portion of the naturalness degradation potential for the xth practice; CLASS: careful logging around small merchantable stems.

**Table 5.** Naturalness degradation potential (NDP) for dead wood (DW) by silvicultural treatments in Quebec's boreal forest.

| Practice | NDP_Factors | % Forest_Area | NDPx |
|---|---|---|---|
| Biomass harvesting | 1 | 0.00% | 0.0000 |
| Thinnings (in natural or plantation) | 0.95 | 2.15% | 0.0205 |
| Plantation—no thinnings | 0.85 | 4.62% | 0.0392 |
| Partial cut | 0.75 | 0.58% | 0.0044 |
| Salvage logging | 0.7 | 0.07% | 0.0005 |
| Careful logging (CL) | 0.65 | 13.93% | 0.0906 |
| Variable retention cut (2% vol) | 0.6 | 0.00% | 0.0000 |
| Undisturbed or natural disturbances | 0 | 78.65% | 0.0000 |
| DW_NDP | | | 0.1551 |
| Actual *DW_PNI* | | | 0.8449 |

Note: NDP_factors: naturalness degradation potential factors related to practices; % for_area: percentage of forested area (in 2012); NDPx: Portion of the naturalness degradation potential for the xth practice.

The naturalness index (NI) calculation then results from the arithmetic mean of the five PNI by characteristic. For the assessment of a given forest management strategy, the calculation should cover a complete harvest cycle (i.e., forest rotation), simultaneously considering the effects over time of the harvest on condition indicators and of silvicultural treatments on pressure measures.

The generic procedure for naturalness assessment (and the corresponding files used for the 3 FMU) is as follows:

1.  Define the territory for which the analysis will be performed and anticipate aggregation of results if the studied area covers multiple data sources.
2.  Identify ecological issues for the studied area based on literature and/or stakeholder consultations.
3.  Pinpoint potential measures available for reference and actual data of the condition and pressure based on literature, forest inventories, and maps.
4.  For each naturalness characteristic, identify condition indicators and corresponding measures which can to be used to assess ecological issues.

5.　For each condition indicator, find a reference value using either historical studies or old forest inventories and maps (for the 3 FMU: SIFORT1 maps (forest information system by tessellation) and Bouchard et al. 2015 [33] for OF).

6.　For each condition indicator, evaluate actual measures using the latest forest inventory map (for the 3 FMU: SIFORT4 maps).

7.　For each condition indicator, set pni curves for the studied area (by changing the reference values for each condition indicator in Table S1) and enter the actual measure to calculate the corresponding pni (by changing the measured values for each condition indicator in Table S1).

8.　For pressure measures, identify the appropriate approach for NDP evaluation related to each naturalness characteristic. Identify curves, set factors based on studies or expert opinion, and get the measures of the area by practice from forest inventory maps (for the 3 FMU: Ecoforest 4 maps, CS_NDP in Table 3, exo_NDP curve in Table S1, and NDP tables (factors and area) for HS, DW, and RP in Table S1).

9.　Calculate the PNI for each naturalness characteristic using Equation (1) (for the 3 FMU: Formulas by characteristic are detailed in Table 2).

10.　Calculate the NI, which corresponds to the arithmetic mean of the five characteristic_PNI.

## 2.2. Test Area

The proposed approach for naturalness assessment was applied to a public forest territory formed by three forest management units (FMU) located in the western black spruce feathermoss bioclimatic sub-domain, near the locality of Chibougamau in Northern Quebec region (Figure 4). These 3 FMU (no 2663, 2665, and 2666) cover a total area of 1,305,200 ha, which is larger than the home range of the boreal caribou, an umbrella species for the boreal forest [34]. Historical data were taken from the first Quebec forest inventory, corresponding to the 1965 to 1974 period, using Quebec's SIFORT system (tessellation of provincial forest inventory maps), from which the 6% of harvested areas and other anthropic disturbances were removed. Current data, corresponding to the 2011 to 2013 period, were taken from the fourth inventory program. The territory used for the analysis covers the whole area included in the perimeter of the FMU (without cutting tessell in SIFORT maps), including the surrounding strict protected areas (IUCN categories I to III) associated with these units. The percentage of forested area over the territory of analysis was calculated for measures of forest condition (CT, LS, OF, and IR) and the percentage of terrestrial area over the territory of analysis for context measures (CF, W_CC, Wm, ANT) was obtained from SIFORT maps (SIFORT1 for "reference" measures, except OF, and SIFORT4 for "actual" measures). Percentages of forested area by origin considering silvicultural treatments in the portion admissible for wood production necessary for weighing NDP_factors were measured with the ecoforest map, which provides polygonal data that are more precise. Each measure was then used to evaluate corresponding pnis or NDPs using curves and tables set for the territory.

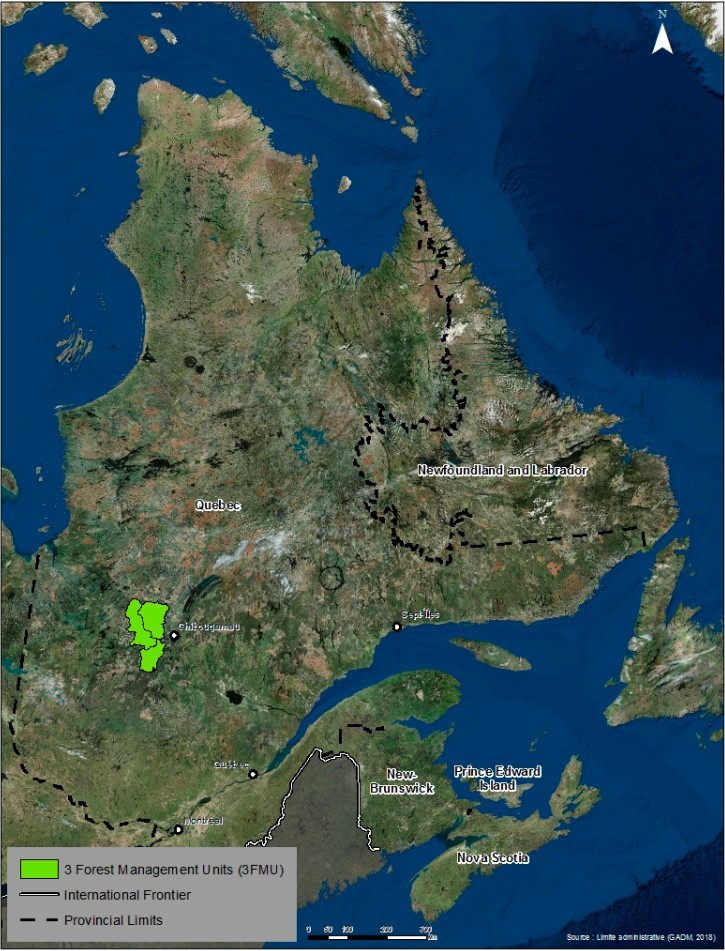

**Figure 4.** Test area localization.

The curves elaborated for pni and NDP evaluation specifically for the 3 FMU are presented in Figure 5 and the NDP factors used in Tables 3–6. Model adaptation to other territories will require calibration of the curves (Figure 5) using appropriate historical/reference values. In our study, all historical values were based on forest inventory data, except for the proportion of forest land covered by old forests (>100 years old), which was based on the modelling study of Bouchard et al. [33]. For landscape context evaluation, the NDP related to clearcut on wetlands was arbitrarily set at 50% of the percentage of wetland area affected by clearcut, as this disturbance was considered less damaging than the drainage of wetlands (NDP_W_CC = 50% × % of wetland area with clearcut). To allow for proper evaluation of extreme scenarios, a two-sided curve was developed for cover type and late successional species groups in order to consider the loss of dominant characteristics on one side, and loss of secondary characteristics on the other. As the reduction of long-lived companion species cannot be measured precisely using inventory data, a reduction factor of 0.2, corresponding to a decrease of one naturalness class, was applied as NDP when diminution was recognized by forest managers; a factor of 0.6 was also tested to evaluate the effect of an hypothetical species extirpation. Improvement of that measure might be possible in the future with more detailed forest composition characterisation performed in more recent forest inventories in Quebec. As dead wood data are not currently available from Quebec's forest inventory, the evaluation was derived from pressure measures resulting from silvicultural treatments, by applying NDP factors weighed by the proportion of forest area by treatments. These factors were estimated based on dead wood data for coarse woody debris measured after a range of silvicultural treatments (careful clearcut logging, plantation, precommercial thinning, biomass harvesting) compared with naturally disturbed forests at the Montmorency Research Forest [35,36].

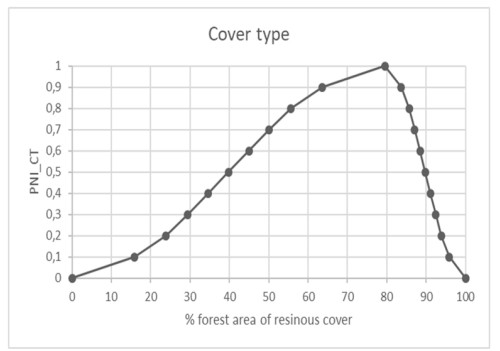

Historical proportion: 79.5% of coniferous cover
Actual proportion: 72.7%
CT_pni = 0.957

(**a**)

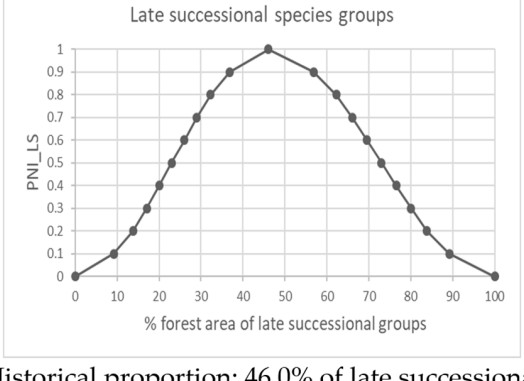

Historical proportion: 46.0% of late successional
Actual proportion: 41.0%
LS_pni = 0.946

(**b**)

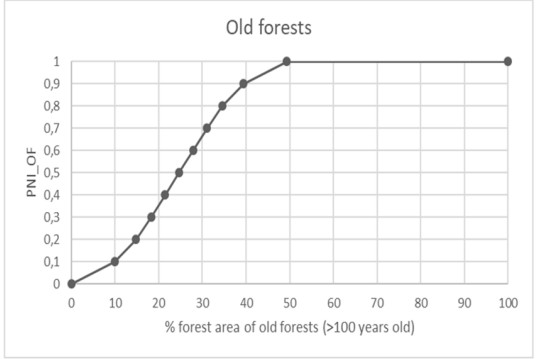

Historical proportion: 49.3% of old forests
Actual proportion: 21.5%
CF_pni = 0.402

(**c**)

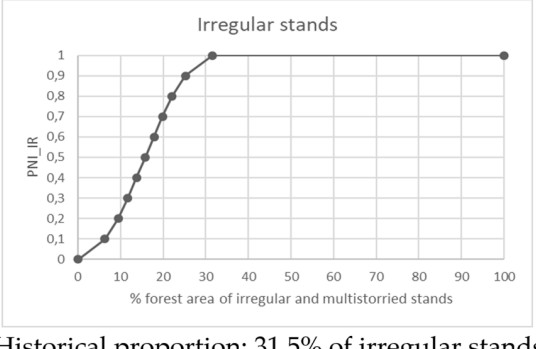

Historical proportion: 31.5% of irregular stands
Actual proportion: 7.0%
IR_pni = 0.121

(**d**)

Historical proportion: 55.5% of closed forests
Actual proportion: 46.1%
CF_pni = 0.919

(**e**)

**Figure 5.** Pni determination curves used for condition indicators' evaluation for three forest management units (3FMU) in the boreal black spruce-feathermoss bioclimatic domain: (**a**) Coniferous cover type; (**b**) late successional species groups; (**c**) old forests; (**d**) irregular stands; (**e**) closed forests.

**Table 6.** Naturalness degradation potential (NDP) for regeneration process (RP) by silvicultural treatments in Quebec's boreal forest.

| Practice | NDP_Factors | % Forest_Area | NDPx |
|---|---|---|---|
| Exotic plantations, afforestation | 1 | 0.00% | 0.0000 |
| Plantation | 0.9 | 5.08% | 0.0458 |
| Seeding | 0.7 | 0.25% | 0.0017 |
| In-fill planting | 0.6 | 0.50% | 0.0030 |
| Salvage logging | 0.55 | 0.07% | 0.0004 |
| Clearcut and final cut | 0.5 | 9.86% | 0.0493 |
| Commercial thinning (natural) | 0.45 | 0.02% | 0.0001 |
| Careful logging (CL) | 0.4 | 5.10% | 0.0204 |
| Partial cut | 0.3 | 0.56% | 0.0017 |
| Undisturbed or natural disturbances | 0 | 78.55% | 0.0000 |
| RP_NDP | | | 0.1224 |
| Actual *RP_PNI* | | | 0.8776 |

Note: NDP_factors: naturalness degradation potential factors related to practices; %for_area: percentage of forested area (in 2012); NDPx: Portion of the naturalness degradation potential for the xth practice.

### 2.3. Description of Tests

Data from 2012 were used to assess the actual naturalness of the 3 FMU. Two series of tests were then performed: (1) Scenario tests on the 3 FMU over time, using three base management scenarios to analyse the sensitivity of the evaluation system to changes in forest management strategies and practices; and (2) hypothetical tests to verify the sensitivity of the model to high pressure levels.

Base management scenarios were: (1) Regeneration through careful clearcut logging on 100% of the harvested area (CL); (2) regeneration through careful clearcut logging on 50% of the harvested area, combined with plantation with thinning on the remaining 50% (CL-PL); and (3) regeneration through plantation on 100% of the harvested area (PL). Careful clearcut logging corresponds to the cut with regeneration and soil protection (CPRS) required by law for clearcut operations in Quebec [37]. For each scenario, the possibility of biomass harvest (bh) over the whole harvested area was also considered [35]. For the plantation, indigenous (PL) or exotic species (PLexo), with a rotation of 70 years for the entire planted portion, were tested. Within the 3 FMU, 17.9% of the forested area have a "strict protected area" status. In order to evaluate its effect on naturalness, the same set of scenarios was applied on the 3 FMU hypothesizing the absence of protected areas. In that case, 95.7% of the forest area would be available for harvest compared to a proportion of 77.8% for the scenario with protected areas. The hypothesis used for the evaluation of naturalness over time for the 3MU are listed in Appendix B. The spreadsheets used for simulation of the 3 FMU through time are provided as Supplementary Materials (Table S2: Area by age class evolution by a 10 year period and Table S3: Composition and irregular evaluation over time) and the procedure for evaluation over time is detailed in Appendix C.

A sensitivity analysis was performed on the 3 FMU CL-PL scenarios (including protected areas), in order to identify the most sensitive variables of the model at time 0, 30, and 70 (corresponding to actual, mid-rotation, and end-rotation): A variation of ±5% was tested independently for each input variable (percentage of area or NPD factor for CS). Scenario results used for this test have been adjusted for exotics, anthropized, and modified wetlands by setting these reference values at 5% to test the influence of a ±5% variation.

Other analyses were carried out to verify the impact of specific assumptions. The effect of the hypothesis used for forest composition after careful logging or plantation was estimated by replacing the COMPO_PNI value at $T_{70}$ by the actual value (COMPO_PNI at $T_0$). A fire cycle of 245 years was used for natural disturbance inclusion in the aging simulation, based on a study located East of the study area [38]. To verify the effect of that factor on the proportions of closed and old forests,

the naturalness at $T_{70}$ was evaluated by setting the fire cycle at 150 years, i.e., the average value for the Western Black spruce-feathermoss domain [39].

An exploratory analysis was also performed to check the model's behaviour after the first rotation. The model was applied for the PL, CL-PL, and CL scenarios with and without strict protected areas beyond the first rotation, up to $T_{150}$, keeping the same hypothesis for composition after CL and PL (as the composition after the second cutting cycle is still not known for these forests).

For the hypothetical extreme tests, eight scenarios were considered (Table 7) with an increasing percentage of exotic species from 0% to 100% of the forest area (0%, 7%, 15%, 30%, 50%, 80%, and 100%). For example, scenario 1 considers 80% of plantation (PL) and 20% of careful logging (CL) along with the use of herbicide in plantations, leading to a coniferous cover of 85% and a proportion of late successional species groups equal to 85% (minus the percentage of exotic species), with a maximal LS value set to 30%. Scenario 2 is similar to scenario 1, but without drainage of wetlands. The hypotheses used for extremes scenarios are listed in Appendix D.

**Table 7.** Extreme scenarios' descriptions.

| Practice or Variable [1] | Sce1 | Sce2 | Sce3 | Sce4 | Sce5 | Sce6 | Sce7 | Sce8 |
|---|---|---|---|---|---|---|---|---|
| Scenario | 80PL | 80PL-0DR | 80PL-0DR-noBH | 10PC | 10PC-0DR-noBH | 100Herb. | 0Herb. | 100PL |
| %PL | 80 | 80 | 80 | 80 | 80 | 80 | 80 | 100 |
| %CL | 20 | 20 | 20 | 10 | 10 | 20 | 20 | 0 |
| %PC | 0 | 0 | 0 | 10 | 10 | 0 | 0 | 0 |
| Herb. | In PL | In PL | In PL | In PL | In PL | All | No | All |
| %DR | 50 | 0 | 0 | 50 | 0 | 50 | 50 | 50 |
| %CT | 85 | 85 | 85 | 85 | 85 | 100 | 60 | 100 |
| %LS | 85-PLexo; ≤30 | id | id | id | id | 85-PLexo; ≤50 | 15 | 100-PLexo; ≤30 |
| CS | Rarefied(R) | R | R | R | R | R | R | Disappeared |
| %OF | 0 | 0 | 0 | 15 | 15 | 0 | 0 | 0 |
| %IR | 0 | 0 | 0 | 10 | 10 | 0 | 0 | 0 |
| BH | Y | Y | N | Y | N | Y | Y | Y |

[1] Sce1: Scenario #1 and so on; PL: plantation; CL: careful logging; PC: partial cutting; DR: drained; CT coniferous cover type; LS: late successional species groups; CS: long lived companion species; OF: old forests; IR: irregular stands; BH: biomass harvest; PLexo: plantation of exotic species; Herb.: use of herbicides; id: idem as preceding.

## 3. Results

Model results for the 3 FMU give a naturalness index (NI) of 0.717 for the year 2012, which corresponds to the near-natural class (Table 8). This naturalness level is explained by the logging of 22.3% of the area, and the rarefaction of some long-lived companion species. The main alteration is related to structure, resulting from the reduction of irregular stands (from 31.5% in the 1970s to less than 7% in 2012) and the loss of old forests (from 49.3% to 21.5% during the same period) relative to the historical state. Detailed results are provided as Supplementary Materials: Table S3 for 3FMU scenarios with protected areas; Table S4 for 3FMU scenarios without protected areas; Table S5 for extremes scenarios.

**Table 8.** Actual (2012) results for the three forest management units.

| Characteristic | PNI | Naturalness Class |
|---|---|---|
| Landscape context | 0.870 | Natural |
| Composition | 0.761 | Near-natural |
| Structure | 0.232 | Altered |
| Dead wood | 0.845 | Natural |
| Regeneration process | 0.877 | Natural |
| Naturalness index (NI) | 0.717 | Near-natural |

### 3.1. Naturalness of Forest Management Scenarios over Time

#### 3.1.1. Naturalness Evolution of the Test Area

Results of the naturalness assessment of the 3 FMU over time are given in Table 9 and illustrated in Figure 6.

**Table 9.** Results of the 3 FMU scenarios over time.

| Scenarios [1] with 17.9% protected areas | | | | | | | | |
|---|---|---|---|---|---|---|---|---|
| Time since 2012 | % logged pro | CL pro | CL-PL pro | PL pro | CL_bh pro | CL-PL_bh pro | PL_bh pro | CL-PLexo_bh pro | PLexo_bh pro |
| 0 | 22.31 | 0.717 | 0.717 | 0.717 | 0.717 | 0.717 | 0.717 | 0.717 | 0.717 |
| 10 | 30.09 | 0.692 | 0.685 | 0.679 | 0.678 | 0.682 | 0.687 | 0.651 | 0.626 |
| 20 | 37.86 | 0.666 | 0.653 | 0.639 | 0.638 | 0.647 | 0.655 | 0.598 | 0.562 |
| 30 | 45.64 | 0.634 | 0.616 | 0.596 | 0.593 | 0.606 | 0.618 | 0.545 | 0.501 |
| 40 | 53.42 | 0.596 | 0.573 | 0.546 | 0.543 | 0.560 | 0.574 | 0.489 | 0.439 |
| 50 | 61.11 | 0.560 | 0.533 | 0.502 | 0.498 | 0.517 | 0.533 | 0.441 | 0.386 |
| 60 | 68.97 | 0.535 | 0.503 | 0.467 | 0.462 | 0.485 | 0.503 | 0.404 | 0.344 |
| 70 | 76.75 | 0.511 | 0.474 | 0.431 | 0.426 | 0.452 | 0.473 | 0.369 | 0.304 |
| Scenarios without protected areas | | | | | | | | |
| Time since 2012 | % logged nopro | CL nopro | CL-PL nopro | PL nopro | CL_bh nopro | CL-PL_bh nopro | PL_bh nopro | CL-PLexo_bh nopro | PLexo_bh nopro |
| 0 | 22.31 | 0.717 | 0.717 | 0.717 | 0.717 | 0.717 | 0.717 | 0.717 | 0.717 |
| 10 | 31.88 | 0.680 | 0.672 | 0.664 | 0.663 | 0.668 | 0.674 | 0.632 | 0.604 |
| 20 | 41.44 | 0.643 | 0.625 | 0.609 | 0.607 | 0.617 | 0.629 | 0.562 | 0.524 |
| 30 | 51.01 | 0.593 | 0.571 | 0.548 | 0.545 | 0.560 | 0.573 | 0.491 | 0.444 |
| 40 | 60.57 | 0.540 | 0.514 | 0.483 | 0.479 | 0.498 | 0.514 | 0.422 | 0.368 |
| 50 | 70.14 | 0.505 | 0.473 | 0.437 | 0.432 | 0.454 | 0.471 | 0.373 | 0.313 |
| 60 | 79.71 | 0.475 | 0.438 | 0.395 | 0.389 | 0.415 | 0.435 | 0.332 | 0.267 |
| 70 | 89.27 | 0.443 | 0.402 | 0.351 | 0.344 | 0.375 | 0.397 | 0.290 | 0.225 |

[1] % logged: cumulative % of forest area regenerated through cutting; CL: 100% Careful logging; CL-PL: 50% careful logging with 50% plantation; PL: 100% plantation; PLexo: plantation of exotic species; bh: biomass harvest; pro: with 17.9% in protected areas; nopro: without protected areas.

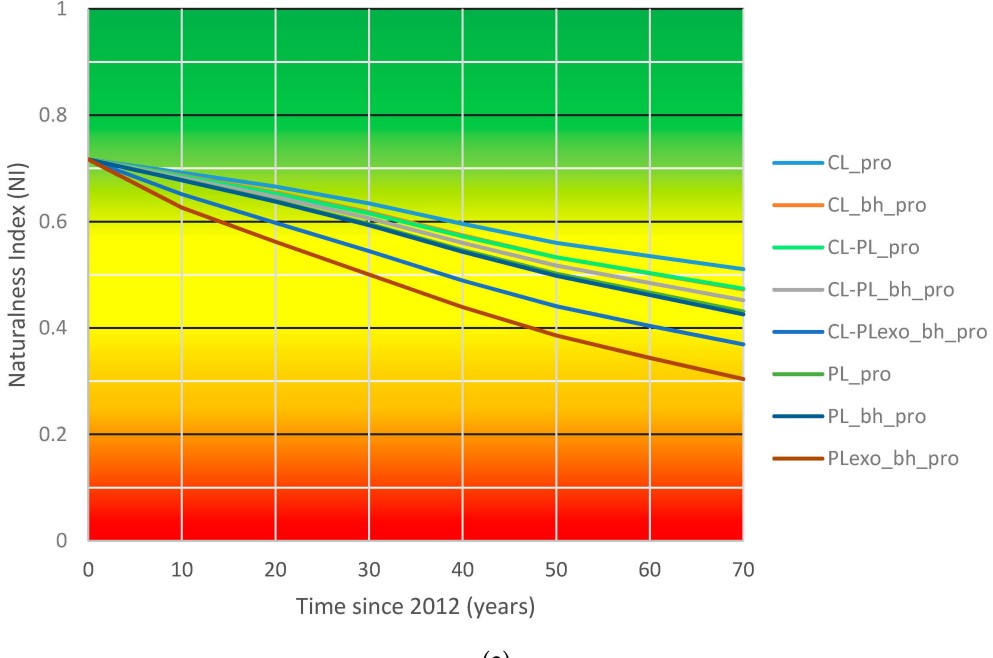

(**a**)

**Figure 6.** *Cont.*

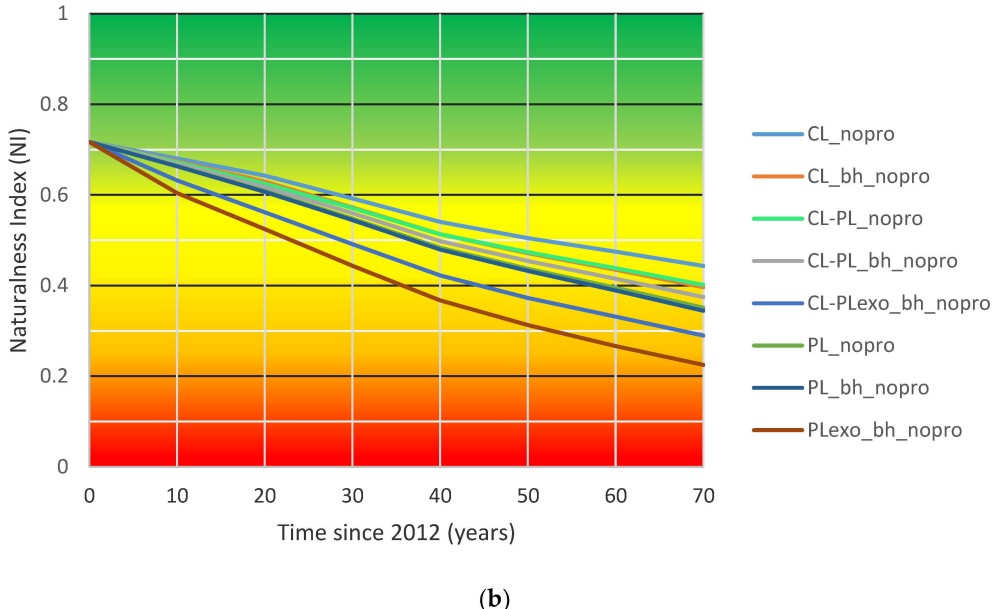

(**b**)

**Figure 6.** Results for the 3 FMU over time: (**a**) With protected areas; (**b**) without protected areas. Scenario description: CL: 100% careful logging, CL-PL: 50% careful logging and 50% plantation of indigenous species, CL-PLexo: 50% careful logging and 50% plantation of exotic species, PL: 100% plantation of indigenous species, PLexo: 100% plantation of exotic species, bh: with biomass harvest, pro: with strict protected areas, nopro: without strict protected areas.

Practicing careful logging only (CL) for the next 70 years on the 3 FMU, taking into account 17.9% of protected areas, would lead to a loss of one naturalness class relative to the current state. With 50% of careful logging and 50% of plantation of indigenous species (CL-PL), which corresponds roughly to the scenario currently applied, the study area would become semi-natural around 2045 and remain in this class for the rest of the period, while it would be from 2040 with 100% plantation of indigenous species (PL) (Figure 6a). In general, for a given ratio of protected areas, the naturalness declines with the proportion of plantation, and more sharply if exotic species are used. After 70 years (which roughly approaches the time required to complete the first cutting cycle), all scenarios that include protected areas would lead the studied 3 FMU to be classified as semi-natural except for those using exotic species, which would lead to an altered state. However, without protected areas (Figure 6b), only two scenarios would lead to a semi-natural class: CL or CL-PL, all the others would be in the altered class (0.4 > NI > 0.2), and the scenario considering 100% of plantation in exotic species over 70 years would be close to the very altered class. After 70 years, the scenario without protected areas corresponds to a rejuvenation of almost 90% of the territory.

After the first cutting cycle, the age structure of the forest would be closer to normalization in the harvested area (i.e., each age class would be more evenly represented among forest stands), and the pressures would cover the whole production area, so the naturalness would stop its decline. Therefore, if ratios per practice and pressures are maintained, and assuming that stand composition following the harvest of the secondary forest would remain unchanged, values for naturalness would tend to stabilize ($NI_{150}$, given on an indicative basis in Tables S3 and S4: with protection: PL = 0.381, CL-PL = 0.444, CL = 0.495; without protection: PL = 0.282, CL-PL = 0.351, CL = 0.406).

With the hypothesis used, forest rejuvenation through 100% careful logging (CL) produces a reduction of the naturalness index over time that is less important ($\Delta NI_{70}$: pro = −0.206; nopro = −0.274) than regeneration through plantation with indigenous species only (PL) ($\Delta NI_{70}$: pro = −0.286; nopro = −0.366). Use of exotic species combined with biomass harvest (PLexo_bh) ($\Delta NI_{70}$: pro = −0.413; nopro = −0.492) would produce around twice as much alteration as natural regeneration through logging itself.

Among the tests performed, it is the application of a forest management regime over the first cutting cycle that has the most important effect on the naturalness index (mean $\Delta NI_{70}$: pro = −0.287; nopro = −0.364). Compared with natural regeneration through CL, the regeneration mode has an important effect when exotic species are used ($\Delta NI_{70}$: pro = −0.207; nopro = −0.218), but a lesser impact when indigenous species are planted over the whole area ($\Delta NI_{70}$: PLpro = −0.08; PLnopro = −0.092; CL-PLpro = −0.037; CL-PLnopro = −0.042). Protection of 17.9% of the forest area has a noticeable effect in limiting the loss of naturalness in the 3FMU (mean $\Delta NI_{70}$ = 0.077).

Biomass harvesting would cause a reduction of the naturalness index of 0.038 and 0.047 after 70 years (Table 9) for the 100% careful logging scenarios with and without protected areas, respectively. This practice has a lesser effect in scenarios with higher levels of plantation: The NI reduction would be 0.005 and 0.007 for the 100% plantation scenario (Table 9) with and without protected areas, respectively, since it is assumed that site preparation prior to plantation would impact the dead wood. With the hypothesis used, the biomass harvest over the entire area would have as much impact on the naturalness as planting indigenous species over 50% of the harvested area. A better evaluation of the effect of the biomass harvest should include small woody debris for DW_NDP factors' evaluation.

When comparing scenarios with and without 17.9% of strict protected areas (Table 9), after 70 years from 2012, the model determines that the CL-PL scenario with protected areas leads to a higher level of naturalness than the scenario with 100% CL without protected areas ($NI_{70}$: 0.474 vs 0.443). The same observation is applied for 100% plantation (PL) with protected areas, which performs better than the CL-PL scenario without protected areas ($NI_{70}$: 0.431 vs 0.402). CL only without protection performs slightly better than PL only with protected areas ($NI_{70}$: 0.443 vs 0.431). The CL-PLexo (plantation of exotic species over 50%) with biomass harvest and protected areas has a higher naturalness than PL of indigenous species over 100% without biomass harvest, but no protected areas ($NI_{70}$: 0.369 vs 0.351). It is important to underline the fact that with protected areas, after 70 years, 77% of the forest area will have been rejuvenated after harvesting, as opposed to 89% for the scenario without protected areas. With protected areas, after 70 years of regeneration through CL, a reduction corresponding to one naturalness class is observed ($NI_0$ = 0.717; $NI_{70}$ CL = 0.511 for a NI loss of 0.206). With plantation of exotic species and biomass harvest, the difference represents more than two classes ($NI_0$ = 0.717; $NI_{70}$ PLexo_bh = 0.304 for an NI loss of 0.413 over 70 years). Without strict protected areas, losses are more important as a larger area is available for cutting.

### 3.1.2. Sensitive Variables and Exploratory Analysis

The results of the sensitivity analysis performed on the 3FMU for the scenario, CL-PL, with strict protected areas for current, 30 year, and 70 year periods are provided in Figure 7. Due to the use of non-linear models, a uniform variation of input parameters (5%) can have a non-linear effect on the results, depending on the curve slope around the parameter value. Therefore, the sensitivity of results can also vary over time (Figure 7). Results proved to be most sensitive to the proportion of forest area covered by exotic species. However, exotic species have never been used in the area under study. Beside the exotics, the most sensitive variables are the percentage of area in irregular (IR), old (OF), and closed forests (CF) at $T_0$; percentage of area in closed (CF), old (OF), and irregular (IR) forests at $T_{30}$; and percentage of area of closed forests (CF), late successional species groups (LS), and modified wetlands (Wm) at $T_{70}$.

As LS is among the most sensitive variables after 70 years, the hypothesis applied for composition after CL or PL might have an important impact on the resulting naturalness index. Assuming no effect on composition (by replacing the COMPO_PNI estimated with the actual value), the $NI_{70}$ would be higher ($\Delta NI_{70}$: PLpro: 0.037; PLnopro: 0.046; CL-PLpro: 0.040 CL-PLnopro: 0.051; CLpro: 0.05; CLnopro: 0.064), in the same range as planting indigenous species over 50% of the harvested area. Therefore, a better assessment of forest composition through time as influenced by silvicultural treatments would be important to improve the reliability of results.

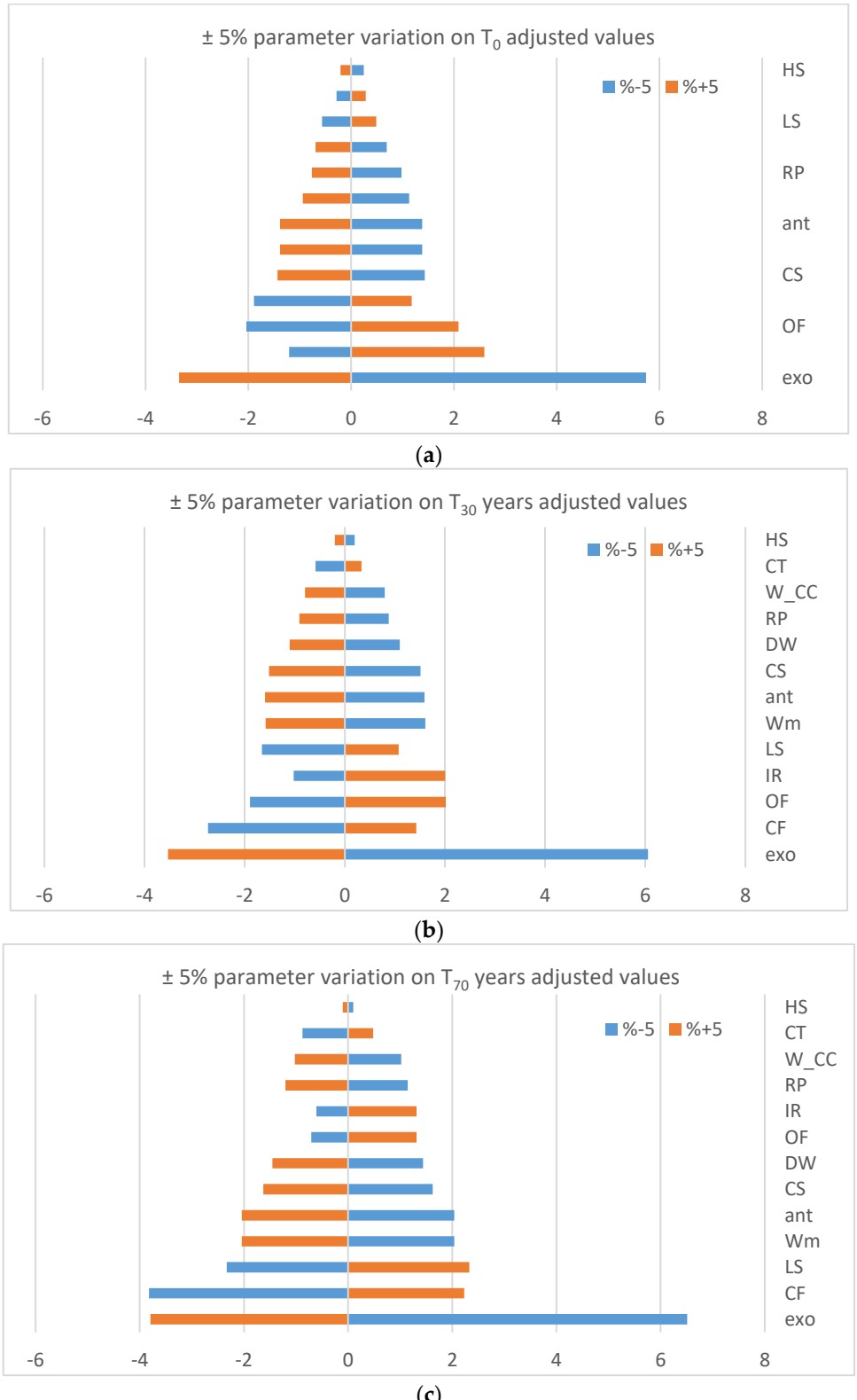

**Figure 7.** Results of the sensitivity analysis testing a variation of 5% of the adjusted parameter value: (**a**) Adjusted values at T0; (**b**) adjusted values at $T_{30}$; (**c**) adjusted values at $T_{70}$; HS: Horizontal structure, CT: coniferous cover type, LS: late successional species groups, W_CC: clear cut on wetland, RP: regeneration process, DW: dead wood, ant: anthropized land; WM: modified wetland, CS: companion species, CF: closed forest, OF: old forest, IR: irregular stands, exo: exotic species.

Considering that a fire cycle of 150 instead of 245 years induces a reduction of the $NI_{70}$ of 0.026 (±0.001) in the scenarios with protected areas and a reduction of 0.030 in the scenarios without protected areas, the model therefore seems relatively robust to the hypothesis used for the fire cycle.

The weight of 0.5 given to estimate the effect on the naturalness of clearcuts on wetlands has a very limited effect considering that clearcuts only affect 7.8% of the wetlands. Hypothesizing 100% instead of 50% would have reduced the resulting $NI_{70}$ by 0.0059, which is marginal.

The results suggest that the model structure, which applies pressure as a reduction of the condition, produces a degradation of the naturalness over the first cutting cycle. The exploratory analysis reveals that beyond that point, results tend to stabilize (see results at $T_{150}$ in Table 10): The strategy is applied over the whole managed territory and therefore the composition after regeneration treatment related to age (here younger and older than 20 years old) becomes constant. The age structure is gradually normalized over the managed area, but continues to evolve with stand aging in the excluded (protected) areas. Applying our hypothesis, we observe a slight reduction of the naturalness between $T_{70}$ and $T_{150}$ in all scenarios.

**Table 10.** Results of the 3 FMU scenarios at $T_{150}$.

| Protection | CL | CL-PL | PL | CL_bh | CL-PL_bh | PL_bh | CL-PLexo_bh | PLexo_bh |
|---|---|---|---|---|---|---|---|---|
| Pro | 0.495 | 0.444 | 0.381 | 0.441 | 0.412 | 0.374 | 0.328 | 0.260 |
| Nopro | 0.406 | 0.351 | 0.282 | 0.339 | 0.313 | 0.272 | 0.231 | 0.164 |

CL: 100% Careful logging; CL-PL: 50% careful logging with 50% plantation; PL: 100% plantation; PLexo: plantation of exotic species; bh: biomass harvest; pro: with 17.9% in protected areas; nopro: without protected areas.

## 3.2. Naturalness of High Pressure Management Scenarios

Extreme scenarios lead to a naturalness index corresponding to altered and very altered classes (Table 11, Figure 8). Higher levels of alteration are associated with an important use of exotic species combined with the loss of companion species. Absence of drainage and, to a lesser extent, application of measures leading to the presence of old forests and irregular stands make it possible to sustain a higher level of naturalness. Scenarios including a homogenous coniferous cover type (scenarios 6 and 8) and a scenario corresponding to a degraded composition with low coniferous cover and late successional representation (scenario 7) lead to a very altered class even without using exotic species. For less extreme combinations, the naturalness class is altered if no exotic species are used, and very altered if a small proportion of exotics is present.

**Table 11.** Extreme scenarios' results. See also Table 7 for a detailed scenario description.

| Variable [1] | Sce1 | Sce2 | Sce3 | Sce4 | Sce5 | Sce6 | Sce7 | Sce8 |
|---|---|---|---|---|---|---|---|---|
| %exotics | 80PL | 80PL-0DR | 80PL-0DR-noBH | 10PC | 10PC-0DR-noBH | 100Herb. | 0Herb. | 100PL |
| 0 | 0.215 | 0.247 | 0.269 | 0,231 | 0.292 | 0.167 | 0.177 | 0.085 |
| 7 | 0.174 | 0.206 | 0.228 | 0,191 | 0.251 | 0.141 | 0.148 | 0.065 |
| 15 | 0.149 | 0.181 | 0.203 | 0.165 | 0.226 | 0.125 | 0.130 | 0.053 |
| 30 | 0.120 | 0.152 | 0.173 | 0.136 | 0.197 | 0.106 | 0.109 | 0.050 |
| 50 | 0.090 | 0.122 | 0.144 | 0.106 | 0.167 | 0.085 | 0.087 | 0.046 |
| 80 | 0.074 | 0.106 | 0.128 | 0.090 | 0.151 | 0.074 | 0.074 | 0.040 |
| 100 | | | | | | | | 0.036 |

[1] SceX: Scenario number, PL: plantation; PC: partial cutting; DR: drained; BH: biomass harvest; Herb.: use of herbicides; Sce1: 80%PL 20%CL Herb. In PL 50%DR CT = 85% LS = (85% − PLexo) ≤ 30 CS = rarefied OF = 0% IR = 0% BH; Sce2: Sce1 without DR; Sce3: Sce1 without DR nor BH; Sce4: Sce1, but 10CL and 10PC so OF = 15% and IR = 10%; Sce5: Sce4 without DR nor BH; Sce6: Sce1, but 100%Herb so CT = 100% and LS = (85% − PLexo) ≤ 50; Sce7: Sce1 without Herb in PL so CT = 60% and LS = 15%; Sce7: 100%PL 100%Herb. 50%DR CT = 100% LS = (100% − PLexo) ≤ 30 CS = disappeared OF = 0% IR = 0% BH.

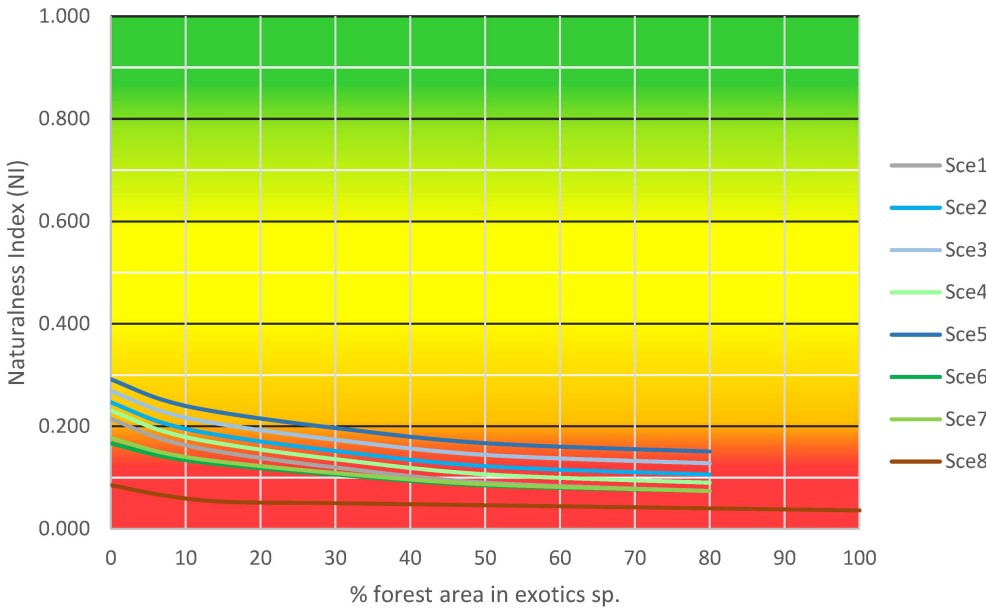

**Figure 8.** Results for the eight extreme scenarios. See also Table 7 for detailed scenario descriptions. Sce1: 80%PL 20%CL Herb. In PL 50%DR CT = 85% LS = (85% − PLexo) ≤ 30 CS = rarefied OF = 0% IR = 0% BH; Sce2: Sce1 without DR; Sce3: Sce1 without DR nor BH; Sce4: Sce1, but 10CL and 10PC so OF = 15% and IR = 10%; Sce5: Sce4 without DR nor BH; Sce6: Sce1, but 100%Herb so CT = 100% and LS = (85% − PLexo) ≤ 50; Sce7: Sce1 without Herb in PL so CT = 60% and LS = 15%; Sce7: 100%PL 100%Herb. 50%DR CT = 100% LS = (100% − PLexo) ≤ 30 CS = disappeared OF = 0% IR = 0% BH.

## 4. Discussion

Decision support systems depend upon summaries and systematic reviews available at the time of their conception and are therefore subject to improvements as new information becomes accessible [26]. Nevertheless, the conceptual model developed in this study provides a basic frame for naturalness assessment, although indicators, measures, and curves might be revised when better information and data become available.

### 4.1. Conceptual Model

Evaluating the intensity of silvicultural management makes it possible to quantify land-use intensity in forests. Gossner et al. [40] showed that biodiversity can be related with land use measures, such as naturalness based on trees species composition, dead wood, and other structural characteristics, or stand management intensity based on tree species, stand age, and aboveground living and dead woody biomass. Therefore, a methodology combining the condition evaluation of composition and structure, and pressure measures resulting from silvicultural practices represents a good proxy for the evaluation of the effects of forestry on biodiversity resulting from a combination of various practices. Further research is needed to verify to which extent the utilization of condition measures can detect improvements of naturalness resulting from restoration or enhanced ecological management strategies. The inclusion of pressure measures can adequately reflect the effects of mitigation measures, although the model should be further validated. Our results suggest that the presence of strict protected areas in a forest landscape compensates, to some extent, for the impacts of intensive management [41].

The model developed in this paper was shown to be sensitive enough to characterize the naturalness of different forestry management systems and therefore discriminate between different wood supplies from a variety of forest management practices. The results demonstrate that the proposed naturalness assessment model can be useful to evaluate the land use intensity of forestry practices at a finer level than existing approaches to inform decision-making in life cycle assessment. For instance, global guidance for life cycle impact assessments [3] currently considers only two levels

of intensity for forestry: Intensive and extensive forestry. For their part, Chaudhary and Brooks [42] proposed to divide secondary vegetation in four classes: Plantations, clearcut, selective logging, and reduced impact logging. In contrast, our approach makes it possible to take into account the condition of the forest as well as the proportion of different practices to more precisely characterize the forest management strategy and its impacts on ecosystem quality.

The application of pressure as a reduction of the condition contributes to an important reduction of the naturalness resulting from the progression of the first cutting cycle over the area. Such an important reduction of the quality observed as a result of the initial land use transformation process is coherent with the conceptual model proposed by LCA developers for ecosystem quality evolution related to land use [43]. However, tests over time highlight some specificities of forestry land use reflected by the model. Contrary to most land uses, the first cutting cycle of a forest land corresponds to a progressive transformation from the natural state to a naturalness level related to the forest management strategy applied. In Quebec's boreal forests, this initial transformation may take up to 100 years. During the subsequent rotations, the naturalness index tends to stabilize as a result of the normalization of the forest, supposing that sustainable management is used. However, our model indicates a trend toward a slow erosion of the ecosystem quality over time during the sustainable production phase. If the land use ever stops and constraints are relaxed (although future land use changes are more likely to progress toward land uses of higher hemeroby), the naturalness should progressively improve as condition indicators will gradually recover with the aging of the forest. However, we do not know if condition indicators will ever come back to the natural state after relaxation. Nevertheless, some pressures will remain (ex: Drained wetlands or other anthropic features, like permanent roads, energy transportation lines, etc.), so theoretically the ecosystem should never recover completely.

Given the model's sensitivity to age related variables, a better integration of plantations would be necessary to improve the results. It would be interesting to explore the application of the naturalness assessment model using data from sustainable harvest calculation systems to reflect the effect of forest management strategy implementation considering simultaneously shorter rotation for plantations, application of the modelled composition and growth, simulation of harvest spatially applied to the admissible area, and a different handling method of natural disturbances [44].

The natural assessment model developed in our study was designed to be easily adapted to other regions using the conceptual model. All five naturalness characteristics should be considered, indicators should be reviewed to include all regionally important ecological issues, and measures should be identified among available data. Curves for condition indicators would then have to be calibrated using the specific historical values of the studied region and NDP factors should be adjusted to reflect regional practices' effects.

Our model could also be integrated in LCA models and used to inform building and construction eco-designers beyond the outcomes of this specific case study. To do so, further work still needs to be done to generate regionalized results across Canadian FMUs and ecosystems. Depending on the availability of historical data, naturalness assessment could be performed for ecological domains or sub-domains (ex. Western black spruce feathermoss sub-domain); aggregated results could be calculated by region or country to allow for the assessment of harvested wood products in a broader context, where the exact provenance of the wood is not known.

*4.2. Naturalness Assessment Application to the 3 FMU*

The results of the case study inform us that wood coming from the 3 FMU has less negative impacts on the quality of ecosystems if the management strategy relies on natural regeneration through careful logging instead of plantation, especially if exotics are involved. It is still possible to limit the potential loss of specialized species resulting from sustainable forest management provided that the proportion of strict protected areas is sufficient to mitigate the degradation of the condition indicators.

The following observations raise questions that should be further addressed in LCA. The actual naturalness of the 3 FMU is near-natural as a result of a rejuvenation through the harvest of 22.3% of

the forest area, including plantation of indigenous species over 5% of the forest area. The difference between management scenarios results from the cumulative effects of practices over time, mainly those inducing rejuvenation, the model being sensitive to age-related variables (OF and CF). Given that the naturalness index tends to stabilize after the completion of the first cutting cycle, the naturalness assessment of a forest management strategy requires an evaluation over the whole cutting cycle. This corresponds also to the potential impact of the forest regime. However, actual naturalness could be used if the objective is to characterize the naturalness of the forest from which the wood is currently procured. The actual naturalness is the result of the practices applied up to now. It does not necessarily correspond to the level resulting from sustainable management, which is more consistent with the evaluation over a whole cutting cycle. In a territory including forests that have never been harvested, such as the 3 FMU, actual naturalness gives an optimistic portrait and does not correspond to the potential impact of the present activities.

Some important limitations of the use of the model in Quebec's boreal forest need to be stressed. The model could not be used to evaluate the naturalness index in the test area beyond the first cutting cycle, as no data was available to describe the future evolution of the composition of secondary forests in that area. Pni and NDP evaluation curves and factors should be validated according to expert opinion.

The uncertainty of the results increases over time; the evolution of forest composition in older secondary forests (>40 years) in this ecological domain has yet to be verified. The future evolution of the natural disturbance regime under a changing climate is also unknown. Therefore, its effects on age structure are unknown as well as the proportion of future regeneration failures, which affects the closed forest coverage. Nevertheless, the importance of the proportion of old forests and irregular stands, and eventually closed forests, is coherent with present concerns related to age structure when attempting to apply ecosystem-based management [27].

## 5. Conclusions

Despite the necessity of further model and parameter validation, the model developed in this paper makes it possible to assess along a single alteration gradient the impact on ecosystem quality of different forestry management systems, simultaneously considering the condition of the forest and the mix of forestry practices involved. Therefore, the model is sensitive enough to differentiate between forest management strategies. The capacity of the model to reach a very altered class was tested with hypothetical high pressure levels associated with the use of exotic species. Tests over time showed that the results are coherent with the conceptual model proposed by LCA developers for ecosystem quality evolution related to land use and highlight some specificities of the forest land use related to forestry. For instance, the initial land use transformation caused by forestry is gradual and the resulting level of naturalness depends upon the management strategy.

The results of this research work set the basis to inform building and construction designers on the potential impact on ecosystem alteration associated to harvested wood products at the landscape level as a function of forest management strategy, considering the condition of the forest and the nature of the adopted forestry practices. The naturalness index will have to be assessed at a regional level and scaled for the wood productivity and eventually aggregated at an upper geographical level in order to be used in life cycle impact assessment methodologies. Whether the naturalness index could be used as a mid-point indicator or is related to the damage category is still an open question and depends on biodiversity data that are available to generate the correlation.

**Supplementary Materials:** The following are available online at http://www.mdpi.com/1999-4907/10/4/325/s1, Table S1: Pni and exo_NDP determination, Table S2: Area by age class evolution by 10 years period, Table S3: Composition and irregular evaluation over time, Table S4: Detailed results over time for the 3FMU with protected areas, Table S5: Detailed results over time for the 3FMU without protected areas, Table S6: Detailed results for the hypothetical extreme scenarios, Data files: Maps: SIFORT1_3MU, SIFORT4_3MU and Ecoforest4_3MU.

**Author Contributions:** Conceptualization, S.C. and L.B.; methodology, S.C. and L.B.; validation, S.C.; formal analysis, S.C.; investigation, S.C.; resources, R.B.; data curation, S.C. and É.T. for dead wood.; writing—original draft preparation, S.C.; writing—review and editing, S.C., L.B., M.M., R.B. and É.T.; visualization, S.C.; supervision, R.B., L.B. and M.M.; project administration, R.B. and M.M.; funding acquisition, M.M.

**Funding:** This research was funded by the Natural Sciences and Engineering Research Council of Canada through a grant to Manuele Margni, grant number CRD-462197-13, with the collaboration of Cecobois, Canadian Wood Council, Desjardins, GIGA, Hydro-Québec and Pomerleau.

**Acknowledgments:** The authors would like to thank M. Denis Chiasson from Barrette Chapais Ltée for providing information related to the studied area, M. David Baril from BFEC for providing links to required information about sustainable yield calculation, Martin Barrette from Forest Research Direction of the Québec's Ministère des Forêts de la Faune et des Parcs for his valuable contribution the concept of Figure 1, and M. Stefano Biondo and his team form Laval University GeoStat Center for their help with maps data handling.

**Conflicts of Interest:** The authors declare no conflict of interest. The funders had no role in the design of the study; in the collection, analyses, or interpretation of data; in the writing of the manuscript, or in the decision to publish the results.

## Appendix A  Naturalness Characteristics, Indicators and Measures

The five naturalness characteristics selected for the naturalness assessment corresponds to (1) landscape context, (2) forest composition, (3) structure, (4) dead wood and (5) regeneration process. Grouping indicator evaluation under naturalness characteristics limit the number of indicators and can facilitate the adaptation of the method to other regions according to available measures and regional ecological issues. Tree species and forest structure are among the most studied traits of naturalness [14]. Compared with others naturalness assessments [14], the naturalness characteristic "Landscape context" has been added to take into account the proportion of forest habitat at the landscape level. Dead wood was distinguished from the structure because forestry practices can have distinct or even opposite effects (ex: biomass harvest impact dead wood directly; partial cut can promote irregular structure but can reduce dead wood amount if harvest reduces mortality). Similarly, old forests and irregular stands are evaluated distinctly because stands over 90 years old are not necessarily irregular. Very old forests generally exhibit irregular vertical structure and important amount of dead wood, but this is not the case for 100 years old stands. The use of three different indicators for old forests (>100 years old), irregular stands and dead wood allows considering the effects of mitigation measures.

This section provides information about indicators and measures used for the naturalness assessment, and the calculation for each characteristic.

*Appendix A.1 Landscape Context*

Landscape context refers to the amount of forest habitat at the landscape level. This characteristic considers closed forest (CF) habitat (density ≥ 25%) over 40 years old as a condition indicator (Table 1), measured using the percentage of terrestrial area of forest stands [38]. This measure evaluates the extent of conditions suitable for forest dependent species, such as certain birds or litter invertebrates [19], or for species needing large undisturbed areas such as the woodland caribou [45]. The measure is sensitive to overabundance of young stands resulting from disturbances and/or regeneration failure following disturbance [46]. Habitat loss has a greater effect on biodiversity than habitat fragmentation [47] to some extent; the latter becomes more important when the amount of habitat represents a low proportion of the landscape, below a critical habitat threshold of 30% [29], at which point many species are lost. Therefore, if the proportion of forested area is equal or below 30%, the inclusion of fragmentation in landscape context evaluation should be considered. However, the model should not be applied if the proportion of anthropized land reaches 70%: such a situation would extend outside of the proposed naturalness gradient.

The pressure on landscape context considers both land use change, identified as one of the main drivers of biodiversity loss by the Millennium Ecosystem Assessment [2], and alteration of wetlands, recognized as an important habitat to protect in the latest Quebec's regulation [48]. The pressure measures are evaluated as the percentage of terrestrial area in anthropic land use, the percentage

of wetlands with an anthropically modified condition (ex: drainage or transmission line) and the percentage of wetlands with clearcut. The resulting *context_PNI* (Table 2) could have a negative value when pressures are severe enough, reflecting an extension of the naturalness gradient into the hemeroby gradient. However, in our model, negative values were set to zero.

*Appendix A.2 Tree Species Composition*

Tree species composition considers two condition indicators: the cover type (CT) (coniferous, deciduous or mixed) which characterizes the forest matrix, and the late successional species groups (LS). Both indicators are measured using percentage of forested area (Table 1). The distribution of forest cover type represents a basic forest characteristic used for forest ecosystem management [49]; it represents an indicator of ecosystem diversity that is essential for biodiversity according to Quebec's forest sustainable management criteria and indicators [50]. The cover type distribution depends on successional status related to the major natural disturbance regime, i.e., fire in Quebec's boreal forest [51]. The percentage of forested area of the dominant forest cover type, namely coniferous for Northeastern America [39], is used as the indicator. This measure allows the detection of a shift toward mixed or deciduous cover types related to forest exploitation, a phenomenon observed in Quebec's boreal forests [52]. In other parts of the world, as in southern Finland, forest management practices such as the use of herbicides reduce the non-dominant cover types (i.e., hardwood and mixed), diminishing tree species diversity [53]. The alteration of natural tree species composition of forest stands results primarily from forest management and past land use, which increase the abundance of stands in their early successional stages [54]. In Quebec's boreal forests, early successional stages include intolerant hardwood species such as trembling aspen (*Populus tremuloides*) and paper birch (*Betula papyrifera*). In the northernmost areas of the boreal forest, the late successional species groups correspond to pure black spruce or old black spruce/balsam fir stands [55]. The increase in extent and frequency of disturbances can lead to a reduction of the abundance of late successional species groups in the landscape. Elsewhere, as in eastern Finland for example, forest management activities can reduce the occurrence of stands in their earlier stages of succession [56].

The pressure measure on forest composition considers exotic species stands (exo), and the decrease of the abundance of long-lived companion species (CS). Even if exotic species are absent from the test area, this indicator has been included to allow model adaptation to other regions and perform the test to high levels of pressure. The NDP from exotic stands is evaluated using the logarithmic curve (Figure 3c). The use of exotic species is considered to have a high potential impact because of the associated risks of genetic pollution or hybridization with indigenous species [57], species naturalization or even invasion [58], and alteration of natural disturbance regime resulting from interaction with disturbance agents such as insects [59]. Higher risks associated with this factor justifies the use of the logarithmic curve. The rarefaction of late successional companion species resulting from forest management activities represents another issue related to forest composition. In Quebec's boreal forests it is the case for eastern white cedar (*Thuya occidentalis*) and white spruce (*Picea glauca*) [27]. Theoretically, high pressure values for both measures applied on the mean condition indicator could yield negative values; to avoid inconsistencies, the minimal value for *compo_PNI* is set to 0 when the calculation leads to negative values.

*Appendix A.3 Structure*

In general, natural stands tend to be structurally heterogeneous, both vertically and horizontally. Structural complexity may determine habitat availability and may thus influence diversity of plant, animal and microbial communities [60]. Relevant measures of forest structure include canopy cover, vertical structure and size or age distribution of trees [61]. The following condition indicators were retained for structure: age structure and vertical structure (Table 1). Horizontal structure was evaluated using a pressure indicator resulting from silvicultural treatments.

Forest age structure corresponds to the relative abundance of stands belonging to different development stages or age classes [33]. In natural forests, the age structure is the result of natural disturbances regimes, mainly driven by fire in Quebec's boreal forest [49]. Disturbance regime modelling of Quebec's boreal forest show that the median proportion of old forests (>100 years old) varies from 49 to 77% [39]. The proportion of forests older than 100 years has never been under 30% in the natural boreal forest [62]. Old forests represent a critical habitat for maintaining biodiversity due to the presence of bigger older trees, internal structural complexity and high abundance of dead wood including large pieces, which are important characteristics for many specialized species [63]. In the crownland managed forest of Quebec, almost 1% of the productive forests is harvested annually [44], reducing the proportion of the old forests [64]. Therefore, the measure used for the condition indicator relates to old forests (OF) and corresponds to the percentage of forest area comprising stands over 100 years old. Vertical structure is measured with the percentage of forest area covered by irregular stands (IR). With age, old forests progressively develop an irregular vertical structure (associated with cohort replacement and gap dynamics due to the death of old trees) along with the presence of old high trees and an abundance of dead wood [55,65]. This structural complexity produces habitat diversification and favors biodiversity, especially among vascular plants, terrestrial mosses, liverworts and lichens [62]. We used two different condition indicators for structure: old forests and irregular stands. 100 years old forests do not necessarily have an irregular structure and the proportion of multi-storied and irregular stands was more important before the onset of the commercial exploitation of the forest. Cohort based management had been proposed to answer this issue and the inclusion of the irregular stands as an indicator allows to consider stands from second and third cohorts [64].

Horizontal structure refers to spatial distribution of stems. Plantation and thinning homogenize the horizontal structure by regularising spatial stem distribution and density [15]. This results in a lower variation within the stand, and therefore a lower variety of microhabitats, suitable for a narrower array of species. Horizontal structure is evaluated through pressure measures resulting from silvicultural treatments using percentage of forest area by treatments multiplied by corresponding NDP factors. The resulting NDP is then applied as a reduction percentage on the mean of the age and vertical structure pni's (Table 2).

*Appendix A.4 Dead Wood*

About one fifth of all forest species are dependent on decaying wood [56]. The relationship between dead wood (DW) and species diversity is higher in boreal forests than in temperate forests [66]. Decaying wood plays a substantial role in many ecological processes [67]: it affects carbon storage, energy flow and nutrient cycles, contributes to the water-holding capacity of the soil, sustains ectomycorrhizal formation and activity and offers a substrate for seedlings establishment. Decaying wood hosts a large number of epixylic bryophytes and lichens, polypores and other decomposer fungi and invertebrates. Coarse woody debris are particularly impacted by wood harvesting [67]. As no dead wood data is available from forest inventory, the PNI evaluation uses pressure related to human intervention associated with specific silvicultural treatments (PNI = 1 − NDP) (Table 5), considering their respective effects on coarse woody debris, weighed by the percentage of forest area affected by each treatment.

*Appendix A.5 Regeneration Process*

Regeneration process (RP) refers to a chain of events necessary to ensure the renewal of the forest and focusses on the method of arrival or persistence of a species on a site during or after disturbance [68]. Regeneration process considers the regeneration mode in fire dependent ecosystems as a benchmark for natural state (*RP_PNI* = 1) and, as for dead wood, applies NDP factors associated with silvicultural treatments, weighed by the proportion of forest area subject to each treatment. The determination of NDP factors (based on expert opinion) compares silvicultural treatments with fire effects considering: seedling provenance (natural or artificial) and their genetic variability; adaptation of regeneration to

fire; regeneration density; protection/destruction of natural advance regeneration; seed tree abundance; effects on forest floor that can be either positive (when necessary to control paludification on sensitive sites) or negative.

**Appendix B  Hypothesis Used for the 3MU Naturalness over Time Evaluation**

- Logging of 1% of the forest area per year corresponding to the forest area included in the area subject to sustainable allowable cut estimation.  The harvest was applied on the oldest age class available.  Scenarios were applied on 10-years time-steps for a total length of 70 years, corresponding roughly to the first cutting cycle, as 22.5% of the forest area has already been harvested once in 2012;
- A rate of 0.408% per year [38] of rejuvenation from natural disturbances was applied by distributing the rejuvenation proportionally to the area of age classes ≥ 30 years old;
- For aging, 2 matrix of area by age classes has been used: rejuvenation from natural disturbances only, and natural disturbances plus harvest (over 1% of the admissible area applied on the oldest age class available at each period).  Then the resulting matrix has been weighed according to the percentage of excluded area from the production forest depending on the scenario: 22.2% for the scenarios with protected areas, and 4.3% without protected areas;
- Anthropic land use, drainage modification and unproductive area were held constant;
- No modification of forest composition resulting from natural disturbance was included.  Composition after logging was based on composition observed on the forest map, distinguishing forest under and over 20 years old: forest under 20 years old: CL: CT = 6.5% coniferous, %LS = 1.1%; PL: CT = 42.8% coniferous, %LS = 0.4%; Forest over 20 years old: CL: CT = 48.1% coniferous, %LS = 13.1%; PL: CT = 86.6% coniferous, %LS = 9.7%.

**Appendix C  Naturalness Assessment over Time Procedure**

1. Starting from actual area distribution by age classes, simulation by time step of aging, natural disturbances effects and rejuvenation from cutting (For the 3FMU: Table S2), to calculate OF and CF for each period;
2. Find hypothesis for composition indicators after silvicultural treatments included in the forest management strategy (For the 3FMU: CT and LS after CL and PL, under and over 20 years old, were compiled from ecoforest maps), calculate CT, LS and IR resulting from practices application for each period (For the 3FMU: Table S3);
3. For each condition indicator, transform the percentage of area computed by period in pni (For the 3FMU: Using Table S1);
4. Considering the proportion gradually rejuvenated as a result of practices application, evaluation of NDP (For the 3FMU: Using NDP appropriate grid or curve for the scenario from Table S1), to calculate exo_NDP, HS_NDP, DW_NDP and RP_NDP;
5. Complete NDP evaluation using hypothesis about their evolution for other pressure measures (For the 3FMU: CS, Wm, W_CC and ANT assumed to be constant);
6. Calculate NI by time step for each scenario (For the 3FMU: Tables S4 and S5).

**Appendix D  Hypothesis Used for Extremes Scenarios**

- These scenarios use a rotation period of 65 years;
- The proportions of unproductive and anthropic lands correspond to those from the 3 FMU;
- No old forests nor irregular stands are considered except when partial cuts are used;
- Recognized reduction of companion species is included.
- Drainage of wetlands is set to 50% when applied.
- No excluded area is considered.

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
