# Peer review of "A Conceptual Model for Forest Naturalness Assessment and Application in Quebec’s Boreal Forest"

_forests, doi:10.3390/f10040325_

Reviewer 1 Report

This manuscript details a method of evaluating the ecosystem impacts of various land management options using naturalness and life cycle analysis concepts.  The manuscript is very well written, detailed, and easy to follow.  The only suggestions I have for improvement are;

1) the express purpose is to inform architects and planners of eco-design buildings on the impacts of forest management strategies on ecosystems.  The manuscript would benefit from more in the discussion or conclusions on how this this group can use the results presented in the manuscript as an aid.

2) I have difficulty with the conclusionary statement "the model developed in this paper makes it possible to adequately characterize the naturalness of forest created by different forestry management systems, simultaneously considering the condition of the forest and the mix of forestry practices involved".  The paper presents a commendable attempt at assessing the future impacts of current forestry practices but, there are so many hypotheticals and assumptions in the work that It's hardly known that it adequately characterizes anything.

3) the quality of the figures needs improvement prior to publication.

Author Response

1) the express purpose is to inform architects and planners of eco-design buildings on the impacts of forest management strategies on ecosystems.  The manuscript would benefit from more in the discussion or conclusions on how this this group can use the results presented in the manuscript as an aid.

One paragraph has been added to the conclusion.

2) I have difficulty with the conclusionary statement "the model developed in this paper makes it possible to adequately characterize the naturalness of forest created by different forestry management systems, simultaneously considering the condition of the forest and the mix of forestry practices involved".  The paper presents a commendable attempt at assessing the future impacts of current forestry practices but, there are so many hypotheticals and assumptions in the work that It's hardly known that it adequately characterizes anything.

The sentence has been change.

3) the quality of the figures needs improvement prior to publication.

Figures 1, 2 and 3 has been insert in JPEG format.

Reviewer 2 Report

Generally, the research is at a high level, but requires some minor improvements described below.

The reviewed article is a part of the currently popular research trend related to the use of indicator methods to assess various phenomena and their use as an element of the decision support system. The topic is very interesting for me and it is a pleasure to read it. The proposed research approach is very good, the method is described in great detail.  The conclusions reflect the results of the research. The naturalness index has been tested in many aspects and can be used more widely in other areas regardless of external conditions.  This proves its universality. Below I present a few detailed remarks which will allow to improve the manuscript:
1) Figure 2 should be changed - its resolution is not very good, which makes it unreadable.
2) When describing the locations of the test area, a figure should be added presenting the location on a national and regional scale. It would also be good to show on the next map the current structure of coverage of the analyzed test area.
3) L430-494 should be shortened by presenting only the most important data - all results are included in the table above the text - it is not necessary to repeat them in such details.
4) In Table 7 not all the symbols are explained below the table (which means Sce1, Sce 2...), in Table 10 there are no columns with the designation %logged, pro, nopro.
(5) The naturalness assessment procedure in Annex B could be part of the description of the method, or it would be useful to prepare a procedure diagram in the description of the test methods.

Author Response

1) Figure 2 should be changed - its resolution is not very good, which makes it unreadable.

Few corrections has been made and insertion has been change to JPEG format

2) When describing the locations of the test area, a figure should be added presenting the location on a national and regional scale. It would also be good to show on the next map the current structure of coverage of the analyzed test area.

A localization map has been added (new Figure 4)

3) L430-494 should be shortened by presenting only the most important data - all results are included in the table above the text - it is not necessary to repeat them in such details.

The part related to the figure has been shortened

4) In Table 7 not all the symbols are explained below the table (which means Sce1, Sce 2...), in Table 10 there are no columns with the designation %logged, pro, nopro.

Table 7 : Explanation of scenario numbering has been added

Table 10 : Table and footnote has been corrected

 (5) The naturalness assessment procedure in Annex B could be part of the description of the method, or it would be useful to prepare a procedure diagram in the description of the test methods.

Appendix B content was moved in the method section.